# Exploring the relationship between frequent internet use and health and social care resource use in a community-based cohort of older adults: an observational study in primary care

Caroline S Clarke,[1] Jeff Round,[2] Stephen Morris,[3] Kalpa Kharicha,[1] John Ford,[4] Jill Manthorpe,[5] Steve Iliffe,[1] Claire Goodman,[6] Kate Walters[1]

► Prepublication history and additional material are available. To view these files please visit the journal online (http://dx.doi.org/10.1136/bmjopen-2017-015839).

For numbered affiliations see end of article.

**Correspondence to**
Dr Caroline S Clarke; caroline.clarke@ucl.ac.uk

## ABSTRACT

**Objectives** Given many countries' ageing populations, policymakers must consider how to mitigate or reduce health problems associated with old age, within budgetary constraints. Evidence of use of digital technology in delaying the onset of illness and reducing healthcare service use is mixed, with no clear consensus as yet. Our aim was to investigate the relationship between frequent internet use and patterns of health or social care resource use in primary care attendees who took part in a study seeking to improve the health of older adults.

**Methods** Participants recruited from primary care, aged >65 and living in semirural or urban areas in the south of England, were followed up at 3 and 6 months after completing a comprehensive questionnaire with personalised feedback on their health and well-being. We performed logistic regression analyses to investigate relationships between frequent internet use and patterns of service use, controlling for confounding factors, and clustering by general practitioner practice. Four categories of service use data were gathered: use of primary National Health Service (NHS) care; secondary NHS care; other community health and social care services; and assistance with washing, shopping and meals.

**Results** Our results show, in this relatively healthy population, a positive relationship (OR 1.72, 95% CI 1.33 to 2.23) between frequent internet use and use of any other community-based health services (physiotherapist, osteopath/chiropractor, dentist, optician/optometrist, counselling service, smoking cessation service, chiropodist/podiatrist, emergency services, other non-specific health services) and no relationship with the other types of care. No causal relationship can be postulated due to the study's design.

**Conclusions** No observed relationship between frequent internet use and primary or secondary care use was found, suggesting that older adults without internet access are not disadvantaged regarding healthcare use. Further research should explore how older people use the internet to access healthcare and the impact on health.

## Strengths and limitations of this study

► Timely study providing an update on older adults' use of the internet at home and on their use of health and other care services.
► Findings on internet use are one aspect of a survey that addressed health and social care resource use, thus being well positioned to capture the everyday experience of community-dwelling older people.
► We cannot speculate on how much internet use was specifically for looking up information on health or accessing health-related services, as opposed to general correspondence, or seeking information on any other non-health services, for example.
► Causality cannot be inferred.
► Small study size (n=454).

## INTRODUCTION

Life expectancy is rising worldwide, and much research seeks to find ways of improving older people's health and well-being.[1] Work has been undertaken by various groups regarding complex interventions designed to alter behaviour to improve health and well-being, enabling older adults to maintain their independence and good health for longer; however, there is no clear consensus on the best approaches.[2–5] It has been argued that the use of technology by older people could help in maintaining health and well-being and/or assist in managing or reducing health-related resource use[6 7]; similarly, other work has suggested that older adults might be disadvantaged if they do not use information and communications technology regularly.[8]

There is significant use of the internet by older people in the UK, particularly by those in their 60s and 70s, but it is not universal and decreases with age (88.3% of 55–64-year-old people had used the internet

**BMJ**

in the last three months in 2016, 74.1% of those in the 65–74 age group and 38.7% of those in the 75+ age group).[9] Research on how older people's use of the internet might influence the way they seek help/use healthcare and other resources is still in its infancy.[10–13] There seem to be differences between how younger and older people use the internet; for example, older adults who use the internet seem to use it primarily for email, whereas a large proportion of younger people use social media sites, both for information and for socialising.[14] As nearly half of older people in Scotland were reported to have multimorbidities with increasingly complex health and other needs,[15] this might influence their use of the internet in relation to their health, as well as there being differences in digital and health literacy compared with younger sections of the population.

In the UK, some general practitioner (GP) practices offer online services to patients, including appointment booking systems, and even online access to patients' own primary care records, although this latter example is not yet widely established. Also, healthcare providers are now assessed and ranked, and patients' opinions regarding services can be found online on the National Health Service (NHS) Choices website. It is not yet clear what the uptake and impact of these various NHS online information resources are across age groups and among other sections of society, but it is conceivable that not using the internet might hinder use of these services and therefore access to healthcare.

Participants in the Well-being Interventions for Social and Health needs (WISH) study[16] were community-dwelling older adults recruited from English primary care settings in the London Borough of Ealing (urban) and Hertfordshire (semirural) and they were sent the Multi-dimensional Risk Appraisal in Older people (MRA-O) as a postal questionnaire. All participants gave informed consent to participate in accordance with ethical guidelines and Good Clinical Practice. The MRA-O is an extension of the Health Risk Appraisal in Older people system,[17–19] including domains identified as having an impact on health and well-being in later life during the Smarter Working in Social and Health Care project.[20 21] Participants were asked questions covering a broad range of health, lifestyle, social and environmental domains, including questions on their use of the internet. The resource use data included information on a wide range of services, both public and privately funded, and data on use of the internet, meaning that this data set could enable us to explore the relationship between internet use and resource use, while considering various possible confounders and adjusting for important covariates.

The aim of this study was to examine the relationship between frequent internet use and different types of health and social care resource use, and to consider whether differences in internet use raise concerns about equity of access and use of care services by older adults.

## METHODS

The methods used in this analysis are compliant with the Strengthening The Reporting of OBservational Studies in Epidemiology guidelines for observational cohort studies.[22]

### Design
Cohort study.

### Participants
A random sample of eligible community-dwelling older adult participants aged ≥65 years from five general practices in two diverse regions of southern England were recruited in 2012 and followed up for 6 months as part of the WISH study.[16] Random sampling was completed by the participating practices using their electronic records systems. Further information on the eligibility criteria for this study is given in previous work.[16]

### Data collection
Potential participants were sent letters by their GPs on behalf of the study group, and 526 of the 1550 contacted in this way responded. Of these, 454 returned the M-RAO. The data collected included physical and mental well-being, functional ability, lifestyle and diet, personal characteristics, loneliness and social networks, use of healthcare and social resources, and internet and mobile phone use. Further details regarding the WISH study recruitment and data collection procedures are described elsewhere.[16]

### Measurements
#### Resource use
The WISH study measured resource use across a range of services, including primary and secondary healthcare, informal and other community healthcare, and support from informal or family carers or social care services. These were captured in this analysis as four individual binary resource use variables, where 'yes' meant that one or more of the difference types of contact listed below had occurred within the last three months:

A. Secondary care: Hospital attendance (A&E, inpatient, outpatient).
B. Primary care: GP/community nurse consultation (by phone, face-to-face, a home visit or a call to NHS Direct).
C. Other healthcare services (either NHS or private): Physiotherapist, osteopath/chiropractor, dentist, optician/optometrist, hearing clinic/audiologist, counsellor, smoking cessation service, chiropodist/podiatrist, emergency services (police, ambulance, fire).
D. Wash/meals: Any paid or unpaid help (e.g., from family member or social care services) with washing, dressing, having a bath/shower, cooking/preparing meals, shopping or meal delivery service. The overall binary variable here returns a 'yes' if any paid or unpaid help was reported.

Participants who responded 'yes' were also asked subquestions in each case, regarding how many contacts

they had had with different services, for example, how many nights the participant stayed in hospital, how many times they spoke with the general practice nurse on the phone. The complete list of questions can be found in the online supplementary appendix. The principal binary questions for the resource variables were used in the analysis instead of counting the numbers of contacts due to high levels of missing data in the subquestions.

### Internet use

The internet use question offered four possible answers: often (most days), sometimes (1–3 days a week), occasionally (less than once a week) and never. For the purposes of this analysis, it was dichotomised as 'often/sometimes' (frequently) versus 'occasionally/never' (infrequently) as the numbers of responses across the four groups were too small to allow meaningful analysis as a four-category variable.

### Covariates

We considered a wide range of patient characteristics for the analyses, including GP practice location type (urban or semirural), season of study entry (summer or autumn), sex, age (in bands: 65–74 years, 75–84 years, 85+ years), ethnicity (White British or other), loneliness status (scoring 0–1 or 2–6 on the de Jong Gierveld six-item short scale[23] corresponds to 'not lonely' or 'lonely', respectively), social isolation status (scoring <12 on the Lubben Social Network Scale corresponds to 'socially isolated'), binary response to 'Do you feel lonely much of the time?', Short Form (SF-12)[24] mental health component summary score (MCS) and physical health component summary score (PCS), occurrence of a recent sudden illness in the three months before baseline, age at which left full-time education (before or after 17 years of age) and receipt of pension (state pension only vs other). GP practice location type was included because patterns of healthcare resource use necessarily vary according to population and practice density. Season of study entry was included as there is evidence that use of healthcare services is seasonal,[25] and the loneliness and social isolation variables were included as there has been some research suggesting that, particularly in older adults, use of healthcare services can sometimes be a substitute for social contact.[26 27] The SF-12 was included as a short quality-of-life measure, and this measure is reported, as is usual, as its two components: MCS and PCS.[24] Pension type and the age at which the participant left full-time education were included as proxy measures for socio-economic status.[28 29] The simplicity of the ethnic group division chosen was due to low participant numbers in any non-White-British group, particularly in the semirural practices.

### Analysis

We undertook panel logistic regression for each of the four dichotomous dependent outcome variables for service use, with the GP surgery contributing random effects. This was included as certain variables could be affected in some way by the GP practice's local policies or working practices, meaning that including these possible effects as random was the most appropriate choice. The covariates for the final multivariate regression models were chosen using the common model selection criteria, the Akaike information criterion (AIC) and the Bayesian information criterion (BIC). Interactions between certain variables were also tested. We report ORs and 95% CIs to investigate the relationship between frequent internet use and different types of resource use, controlling for patient characteristics described above. The age group variable was included as a factor variable to remove the assumption of a linear effect with age. Its joint significance was also tested using the $\chi^2$ test. The data were set in Stata v14 as panel data using the patient ID code as the panel variable, and the number of months' follow-up was set as the time variable (0, 3 and 6 months), although exclusion of the time variable when setting the data led to no difference in the regression results.

### Missing data

Demographic data were completed by all 454 participants who returned the M-RAO, except for 7 missing responses to the ethnicity question. Other questions and subquestions were not always completed. We used complete case analysis for the four panel regression models and have not imputed any missing data. Numbers of missing data in each case are detailed in the tables below, with the largest proportion of missing data at baseline being 11% (50/454) in the de Jong Gierveld loneliness variable. Most variables in these analyses had much lower proportions of missing data (~2%). With such low rates of missing data, it was decided that undertaking multiple imputation to estimate new values would not be an efficient use of time. At later time points, there were some dropouts, leading to 89% retention at the 3-month time point and 77% retention at 6 months.

## RESULTS

### Sample baseline characteristics

Fixed patient characteristics measured at baseline for those covariates used in the final models, for the overall group and split by internet use, are given in table 1. There was a large amount of missing data in the subquestions regarding numbers of each specific type of contact in each of the four resource use types, with between 3.4% and 48.1% of those who responded 'yes' to the principal question failing to then state any numbers of contacts. Participants' use of the internet was asked as a four-category question: often (most days) (44%); sometimes (1–3 days a week) (11%); occasionally (less than once a week) (8%); and never (37%), and this was dichotomised as frequently (55%) and infrequently (45%) in the analysis.

### Retention at later time points

The total number of participants in the WISH study at baseline was 454, dropping to 405 (89% retention) at the 3-month time point and 348 (77% retention) at 6 months. The resource use variables that were recorded

**Table 1** Baseline characteristics of participants, given for the overall group, as well as split according to whether or not they used the internet frequently

| Covariates | Overall (n=454) | Using internet infrequently (n=198) | Using internet frequently (n=247) |
|---|---|---|---|
| **Site (semirural; other option was urban) (%)** | **62.3** | **63.6** | **62.4** |
| Season at start (autumn; other option was summer) (%) | 47.6 | 47.5 | 48.2 |
| Gender (female) (%) | 52.9 | 58.6 | 48.6 |
| *Age bands (years, %)* | | | |
| 65–74 | 59.9 | 44.4 | 73.7 |
| 75–84 | 33.3 | 42.4 | 24.3 |
| 85+ | 6.8 | 13.1 | 2.0 |
| White British (seven missing) (%) | 86.1 | 84.4 | 87.4 |
| Lonely (six-item de Jong Gierveld score) (50 missing) (%) | 34.9 | 38.7 | 31.2 |
| Short Form-12 mental score (mean, SD) (42 missing) | 53.2, 8.6 | 52.6, 8.2 | 53.7, 8.9 |
| Short form-12 physical score (mean, SD) (42 missing) | 43.9, 12.5 | 40.6, 12.7 | 46.5, 11.7 |
| Recent sudden illness (12 missing) (%) | 17.0 | 18.2 | 15.7 |
| Left full-time education before 17 years of age (three missing) (%) | 60.8 | 74.0 | 49.6 |

at each time point and form the panel data set used in this analysis showed low proportions of missing values, such that only ≤4% participants were excluded from the complete case analyses on the basis of missing resource use data (see table 2).

**Univariable unadjusted analyses**
Shown in table 3 are the raw unadjusted relationships between each of the covariates included as confounders in the final multivariable models and each binary service use variable. These results show the relationship between each resource use variable and each covariate, with no controlling for any other covariate.

**Multivariable adjusted analyses**
Models with controlling variables included were constructed using the AIC and BIC, and gave an improved fit to the data compared with the univariable models. The controlling variables included were age, sex, site, season at start, SF-12 MCS and PCS, having had a recent sudden illness, ethnicity, age at which left full-time education and de Jong Gierveld loneliness status. Interactions between

**Table 2** Number of respondents at each time point (baseline, 3 months and 6 months), proportions of participants using each type of service at each of the three time points in the panel data set and numbers of missing values

| Resource use variable | Baseline (n=454) | 3 months (n=405) | 6 months (n=348) |
|---|---|---|---|
| *(A) Secondary care (%)* | | | |
| Yes | 37 | 40 | 35 |
| No | 63 | 60 | 65 |
| | 10 missing | 2 missing | 14 missing |
| *(B) Primary care  (%)* | | | |
| Yes | 74 | 75 | 71 |
| No | 26 | 25 | 29 |
| | 11 missing | 6 missing | 15 missing |
| *(C) Other healthcare (%)* | | | |
| Yes | 54 | 57 | 50 |
| No | 46 | 43 | 50 |
| | 11 missing | 3 missing | 13 missing |
| *(D) Wash/meals  (%)* | | | |
| Yes | 14 | 18 | 15 |
| No | 86 | 82 | 85 |
| | 8 missing | 2 missing | 12 missing |

**Table 3** OR (95% CI) from univariable unadjusted analyses of each individual covariate and its relationship to each resource use variable in the panel data set

| Independent variables | (A) Hospital use | (B) Primary care use | (C) Other healthcare services | (D) Wash/meals assistance |
|---|---|---|---|---|
| Frequent internet use | 0.76 (0.47 to 1.25) | 0.79 (0.52 to 1.20) | 1.42 (0.97 to 2.09) | 0.12*** (0.05 to 0.29) |
| *Age bands (years)* | | | | |
| 65–74 | Reference case | Reference case | Reference case | Reference case |
| 75–84 | 1.44 (0.85 to 2.44) | 1.54 (0.98 to 2.42) | 0.94 (0.62 to 1.42) | 7.36*** (2.98 to 18.19) |
| 85+ | 1.55 (0.59 to 4.08) | 4.76*** (1.70 to 13.36) | 1.50 (0.68 to 3.34) | 176.38*** (33.87 to 918.41) |
| Gender (male) | 1.13 (0.69 to 1.84) | 0.95 (0.62 to 1.43) | 0.57** (0.39 to 0.83) | 0.60 (0.25 to 1.41) |
| Site (urban) | 1.35 (0.82 to 2.23) | 0.95 (0.62 to 1.46) | 1.40 (0.94 to 2.08) | 2.76* (1.15 to 6.63) |
| Season at start (autumn) | 0.71 (0.44 to 1.16) | 0.71 (0.47 to 1.07) | 0.84 (0.57 to 1.24) | 2.26 (0.95 to 5.37) |
| SF–12 mental score | 0.97 (0.94 to 1.00) | 0.96** (0.93 to 0.98) | 0.97* (0.94 to 0.99) | 0.94* (0.89 to 0.99) |
| SF–12 physical score | 0.94*** (0.92 to 0.96) | 0.95*** (0.93 to 0.97) | 0.98** (0.96 to 0.99) | 0.85*** (0.82 to 0.89) |
| Recent sudden illness | 5.07*** (2.68 to 9.59) | 3.40*** (1.79 to 6.46) | 3.46*** (2.00 to 5.99) | 4.69** (1.54 to 14.30) |
| Not White British | 0.84 (0.41 to 1.73) | 1.31 (0.70 to 2.47) | 0.77 (0.44 to 1.35) | 2.79 (0.82 to 9.48) |
| Left full-time education aged ≥17 | 1.21 (0.73 to 1.99) | 1.10 (0.72 to 1.68) | 2.29*** (1.55 to 3.40) | 1.22 (0.51 to 2.92) |
| Lonely (six-item de Jong Gierveld) | 1.79* (1.05 to 3.05) | 1.71* (1.08 to 2.72) | 1.65* (1.07 to 2.52) | 5.41*** (2.12 to 13.79) |

*p<0.05; ** p<0.01; *** p<0.001.

pension and internet use, between age at which left full-time education and internet use, and between binary social isolation variable derived from the Lubben Social Network Scale and binary response to 'Do you feel lonely much of the time' were tested, but did not improve the model fit for any of the four regressions and so were not included. The multivariable models' results are shown in table 4 and outlined here below.

### Hospital use

When controlling for age, sex, site, season at start, SF-12 MCS and PCS, having had a recent sudden illness, ethnicity, age at which left full-time education and loneliness, there was no observed association between hospital use and frequent internet use (OR 0.98, 95% CI 0.25 to 3.84) (see table 4).

### Primary care

Use of primary care services, controlling for all the same variables, was also not associated with frequent internet use (OR 1.15, 95% CI 0.78 to 1.70) (see table 4).

### Other healthcare

Frequent internet use was, however, positively associated with use of other healthcare services (e.g., optician, dentist, physiotherapist), when controlling for all the same variables (OR 1.72, 95% CI 1.33 to 2.23) (see table 4).

### Washing/meals assistance

Of those participants who stated that they were using assistance of this nature, approximately a quarter were paying for these services. Receipt of assistance (paid or unpaid) for washing, cooking and similar tasks was not associated with frequent internet use when controlling for all the same variables (OR 0.56, 95% CI 0.12 to 2.55) (see table 4).

### DISCUSSION

Our results show that, in this relatively healthy older adult population, there was a strong and positive relationship between frequent internet use and use of any community-based health services such as physiotherapist, osteopath/chiropractor, dentist, optician/optometrist, hearing clinic/audiologist, counsellor, smoking cessation service, chiropodist/podiatrist and calls to the emergency services (see online supplementary appendix). Use of the internet could be implicated in a person's ability to find any of these community-based services, except perhaps the emergency services. It is not possible to infer a causal relationship between frequent internet use and community health service use based on this analysis. The relationship could have arisen due to one of the following reasons: participants using the internet in order to research services that they wish to use or participants using services being influenced by other service users or other associated factors and thereby encouraged to use the internet. However, there could equally be no relationship at all as correlation does not imply

**Table 4** OR (95% CI) from multivariable adjusted regression analyses for the four final models: one for each type of resource use in the panel data set

| Four models → | (A) Hospital use | (B) Primary care use | (C) Other healthcare services | (D) Wash/meals assistance |
|---|---|---|---|---|
| Dependent variable: frequent internet use | 0.98 (0.25 to 3.84) | 1.15 (0.78 to 1.70) | 1.72*** (1.33 to 2.23) | 0.56 (0.12 to 2.55) |
| *Age bands (years)* | | | | |
| 65–74 | Reference case | Reference case | Reference case | Reference case |
| 75–84 | 1.50 (0.79 to 2.85) | 1.35 (0.92 to 1.98) | 1.08 (0.57 to 2.07) | 3.44*** (1.69 to 6.97) |
| 85+ | 0.60 (0.15 to 2.44) | 2.01 (0.54 to 7.41) | 1.15 (0.45 to 2.97) | 19.36*** (8.21 to 45.64) |
| Gender (male) | 1.68* (1.01 to 2.81) | 0.97 (0.68 to 1.39) | 0.59*** (0.50 to 0.70) | 0.69 (0.24 to 1.95) |
| Site (urban) | 1.47* (1.07 to 2.04) | 0.75 (0.37 to 1.54) | 1.24 (0.84 to 1.84) | 1.46 (0.60 to 3.52) |
| Season at start (autumn) | 0.86 (0.71 to 1.05) | 0.73* (0.56 to 0.95) | 0.77 (0.57 to 1.04) | 1.46 (0.81 to 2.63) |
| Short form-12 mental score | 1.01 (0.99 to 1.03) | 0.96*** (0.95 to 0.98) | 0.99 (0.94 to 1.04) | 0.95 (0.91 to 1.00) |
| Short form-12 physical score | 0.94*** (0.92 to 0.96) | 0 (0.93 to 0.98) | 0.97* (0.95 to 1.00) | 0.87*** (0.84 to 0.91) |
| Recent sudden illness | 4.80*** (2.46 to 9.41) | 1.89** (1.22 to 2.94) | 2.27** (1.27 to 4.07) | 1.42 (0.75 to 2.71) |
| Not White British | 0.43** (0.25 to 0.75) | 1.28 (0.51 to 3.18) | 0.42** (0.24 to 0.72) | 0.69 (0.16 to 2.95) |
| Left full-time education aged ≥17 | 1.68*** (1.35 to 2.09) | 1.23 (0.78 to 1.94) | 2.72*** (2.19 to 3.39) | 3.19*** (2.31 to 6.63) |
| Lonely (six-item de Jong Gierveld) | 1.42 (0.80 to 2.51) | 0.99 (0.64 to 1.54) | 1.08 (0.56 to 2.06) | 1.14 (0.54 to 2.39) |

*p<0.05; ** p<0.01; *** p<0.001.

causation: those interested in and capable of using the internet might simply also prefer to use the services that are on offer.

We did not observe disadvantages in terms of accessing primary or secondary healthcare in those who used the internet infrequently, although the study was not powered to detect such differences. We also did not observe a disadvantage in accessing informal assistance with washing and meals. This is perhaps surprising as needing assistance with washing and meals suggests significant impairment in functioning, which might also impact on internet use. No firm conclusions can be drawn, however, as we do not know from the study what the internet use entailed; for example, if participants used the internet to find out information about their health or local health and care services, or for other reasons.

Our analysis explored the situation regarding access to services that are not currently restricted to online-only access. However, some services in healthcare and other industries are moving towards being offered only online, and a report by Age UK[30] discusses this move towards online-only services, noting that older people and other digitally unengaged groups could potentially be left behind if they are not online. This is an important aspect to the future accessing of healthcare services that we have not been able to address in our analysis.

Notably, there are various initiatives under way to increase the online presence and activity of GP practices,[31] and some concerns have been raised that this might disadvantage those who use the internet less frequently, for example, some older adults, particularly women aged >75, or other disadvantaged groups such as those with disabilities.[9] On the other hand, it has also been postulated that use of online GP services by younger or more technologically literate patients frees up time for receptionists to respond to older adults' telephone calls.[32] Our results are consistent with preliminary suggestions that there might be no cause for concern regarding increasing inequity of access for older people as a whole in the current context, though there may be smaller subgroups within this population who are adversely affected. This present study lacked sufficient power to confirm or refute this, and our patient group was a relatively healthy group, recruited via primary care.

Several factors can contribute to the digital divide between older and younger age groups. These can include a lack of infrastructure, that is, lack of access to broadband and/or Wi-Fi, as well as individual difficulties with learning how to use the internet for those who are acquiring these skills in later life.[33] It is also thought that, besides differences in digital and health literacy compared with younger sections of the population, some older people's complex comorbidities and other needs might also influence their use of the internet in relation to their health.[15] Research on how older people's use of the internet might influence the way they seek help or use healthcare and other resources is however still in its infancy.[10–13]

Statistics published by the Office for National Statistics[9] state that levels of internet use are growing, and the proportion of adults who had either never used the internet or not used it in the last three months had decreased by 13.3 percentage points since 2011. Women aged >75 years have undergone the largest rise in 'recent' (i.e., having used the internet in the last three months) internet use since 2011, although less than a third of this group (32.6%) were recent users in 2016. People aged >75 years consistently have the lowest internet usage rates, in agreement with our observations, but these rates are increasing: from 19.9% of this age group in 2011, to 33.0% in 2015, and 38.7% in 2016.[9] These figures suggest that the digital divide between younger and older age groups might be diminishing in terms of a simple measure of internet use.

Similarly, a report by Age UK[34] suggested that the numbers of older adults using the internet have grown such that now more people aged >65 years have used the internet at some point in their life than have not. It is possible however that the speed at which older adults take up effective use of the internet will be slower than the speed at which some services progress to online-only access, so healthcare services and other industries must take care not to restrict access along these lines if they do not wish to disadvantage older adults and other digitally unengaged groups.

Limitations of this analysis are that the sample size is relatively small, and that resource use was binary, rather than counting the number of contacts that participants had made (this was due to missing responses to subquestions regarding the numbers of specific contacts). In addition, no causality can be inferred due to the nature of the study, and we do not have comprehensive information on the reasons for participants' internet use. We cannot speculate on how much of their internet use was specifically for looking up information on health, as opposed to keeping in touch with family and friends, or obtaining information on transport services or tradespeople, for example. The population that took part in this study has been compared with 2011 census data, and the study population was slightly younger, more likely to be owner-occupiers and less likely to be in an ethnic minority than the census population.[16] The representativeness of the sample is also limited by the low questionnaire return rate of those approached via the initial letter from the GP, which meant that 29% of those initially approached chose to take part and returned the completed questionnaire at baseline.

## Implications

This is one of only a few studies that has investigated internet use alongside the use of other services. Our findings were exploratory and suggest the need for further research to better understand the relationships. In the future, in order to obtain more precise information on the nature of the relationship between technology use and use of health or social care services, further detail could be asked regarding the purpose of internet use, actions taken as a result of internet access and what type of device is used to access the internet. Online technology changes very quickly, and this study offers a timely update on its use by older people living at home. Future work should aim more to understand how older people use technology for their own healthcare both in terms of content and as a way to access information. The use of the internet by older people in long-term care facilities and in hospitals remains underexplored.

**Author affiliations**
[1]Research Department of Primary Care & Population Health, University College London (UCL), London, UK
[2]School of Social and Community Medicine, University of Bristol, Bristol, UK
[3]Department of Applied Health Research, University College London (UCL), London, UK
[4]Department of Public Health and Primary Care, Norwich Medical School, University of East Anglia, Norwich, UK
[5]Social Care Workforce Research Unit, King's College London, London, UK
[6]Centre for Research in Primary and Community Care, University of Hertfordshire, Hatfield, UK

**Contributors** CSC, JR and SM designed the analysis, with important intellectual input from KK, JF, JM, SI, CG and KW. CSC cleaned the data set with assistance from KK. CSC conducted the analysis and prepared the manuscript draft. All authors gave valuable input during the drafting process and approved the final manuscript.

**Funding** The WISH study was funded by the Medical Research Council (MRC) LLHW G1001822/1.

**Disclaimer** The MRC had no role in the design, collection, analysis or interpretation of data; in the writing of this manuscript; or in the decision to submit the manuscript for publication.

**Competing interests** None declared.

**Patient consent** Obtained.

**Ethics approval** Ethical approval for the WISH study was granted by London-East Research Ethics Committee (reference 11/LO/1814) which included permissions to conduct the analysis reported in this paper.

**Provenance and peer review** Not commissioned; externally peer reviewed.

**Data sharing statement** No additional data available.

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
