## [Reviewer comments · BMJ Open]

ARTICLE DETAILS

TITLE (PROVISIONAL)	Exploring the relationship between frequent internet use and health and social care resource use in a community-based cohort of older adults: an observational study in primary care
AUTHORS	Clarke, Caroline; Round, Jeff; Morris, Stephen; Kharicha, Kalpa; Ford, John; Manthorpe, Jill; Iliffe, Steve; Goodman, Claire; Walters, Kate

VERSION 1 - REVIEW

REVIEWER	Dr Pippa Burns University of Wollongong, Australia
REVIEW RETURNED	26-Jan-2017

GENERAL COMMENTS	This is a well written paper looking at internet use and health care service access amongst older adults. However, the paper is limited by the study design, which is acknowledged by the authors. The data is from a larger survey looking at the feasibility of implementing a risk-appraisal system. However, the survey tool failed to ask respondents how they use the internet e.g. for email, travel or health information etc. It also failed to ask respondents how they accessed the internet e.g. computer, i-pad, or smart phone; information that is potentially useful in the development of online information portals and new resources targeting older adults. These oversights severely limit its contribution to our understanding of internet use by older adults. The methodology section should be strengthened by inclusion of information on how the participants were identified and recruited to the study. There is a lack of discussion regarding the impact of known demographic variables on internet access. This is important, particularly in relation to the discussion relating to the drive to increase the presence and activity of GP practices online.
--

REVIEWER	Y. Alicia Hong Texas A&M University, College Station, Texas, USA
REVIEW RETURNED	31-Jan-2017

GENERAL COMMENTS	This paper reports a cross-sectional analysis of Internet use and use of health services in older primary care attendees. Given that data on older adults' (aged >65) Internet use were limited, the study has its merit and significance. Major comments: The selection of covariates needs to consider possible confounding
--

	relationships. The objective of this study was to investigate the relationship between frequent Internet use and use of different types of services. Literature has suggested that Internet use is an important indicator of socio-economic status (SES), or digital divide mirrors the gaps in SES. The apparent relationship between Internet use and health service use was actually the relationship between SES and health service use. The data showed that frequent Internet use was significantly associated with use of specialized health services (therapist, dentist, etc.) but no relationship with primary or secondary care. The data were consistent with literature, i.e., people with higher SES are more likely to use specialized services; and people of low SES only use basic health services. Your study did not measure SES (education, income, welfare status) and therefore could not address the important confounding effect. For any relationship study, the selection of covariates was determined by the study purpose, and each covariate needs clear rationale for being included in the model. Most covariates were included without any explanation, for example, loneliness scale, season at start. Further, why SF24 was used and why it was broken down to two scores? One of the outcome variables was “wash/meals assistance”, an important indicator of disability. SF-12 physical score is a good measure of physical health or disability, so no surprise that it’s a strong predictor of using wash/meals assistance. Therefore to examine the relationship between frequent use and “wash/meals assistance” is less meaningful without explaining the relationship between disability and wash/meals assistance. Other comments: The Introduction should go straight to the purpose of the study and build up the significance and rationale of the study. The Introduction has limited literature on digital divide and health disparities but includes lots of information about MRA-O and WISH, it appears the study was driven by the data from WISH not the other way around. The Methods should follow the format of published articles and detail data sources, sampling and collection methods. If scales were used, Cronbach alpha should be reported. The Data Analysis section should describe whether hierarchical modelling was used given that data were collected from multi-stage sampling strategy and longitudinal design. Why GP surgery contribution was used as random effect needs explanation too. How missing data were handled needs to be included in Methods. If only complete cases were included in data analysis, the actual sample size should be reported. Results: the tables should be formatted following examples of published articles; usually % is sufficient when describing sample characteristics. When reporting relationship (either binary or multivariate) it’s a common practice to indicate p-value. Discussion: it’s important to discuss the study within the existing literature, especially related studies on older adult’s digital divide and health disparities.
--	--

VERSION 1 – AUTHOR RESPONSE

Reviewer(s)' Comments to Author:

Reviewer: 1
 Burns, Pippa
 Univ Wollongong

Please leave your comments for the authors below

This is a well written paper looking at internet use and health care service access amongst older adults. However, the paper is limited by the study design, which is acknowledged by the authors. The data is from a larger survey looking at the feasibility of implementing a risk-appraisal system. However, the survey tool failed to ask respondents how they use the internet e.g. for email, travel or health information etc. It also failed to ask respondents how they accessed the internet e.g. computer, i-pad, or smart phone; information that is potentially useful in the development of online information portals and new resources targeting older adults. These oversights severely limit its contribution to our understanding of internet use by older adults.

- We acknowledge these limitations and thank the reviewer for their comments. This paper contributes to the growing evidence that older people are increasingly using the internet and the implications for communications with primary health care services. It was not the aim of our analysis to develop online information portals or new resources, more to understand how access to the internet may influence health outcomes and service use in older people.

The methodology section should be strengthened by inclusion of information on how the participants were identified and recruited to the study.

- We thank the reviewer for this suggestion, and have added information on identification and recruitment to the paper (page 5, Data collection paragraph).

There is a lack of discussion regarding the impact of known demographic variables on internet access. This is important, particularly in relation to the discussion relating to the drive to increase the presence and activity of GP practices online.

- We agree that this is an important discussion point, and have made additions in the Discussion relating to these points (see page 12, last two paragraphs; and page 13, first paragraph).

Reviewer: 2
Hong , YA
Texas A and M University System

Please leave your comments for the authors below

This paper reports a cross-sectional analysis of Internet use and use of health services in older primary care attendees. Given that data on older adults' (aged >65) Internet use were limited, the study has its merit and significance.

- We thank the reviewer for this observation.

Major comments:

The selection of covariates needs to consider possible confounding relationships. The objective of this study was to investigate the relationship between frequent Internet use and use of different types of services. Literature has suggested that Internet use is an important indicator of socio-economic status (SES), or digital divide mirrors the gaps in SES. The apparent relationship between Internet use and health service use was actually the relationship between SES and health service use. The data showed that frequent Internet use was significantly associated with use of specialized health services (therapist, dentist, etc.) but no relationship with primary or secondary care. The data were consistent with literature, i.e., people with higher SES are more likely to use specialized services; and people of

low SES only use basic health services. Your study did not measure SES (education, income, welfare status) and therefore could not address the important confounding effect.

- We thank the reviewer for these comments. We used “age at which left full-time education” as a proxy for SES (before 17 years of age = low SES; 17 or older = high SES), as has been done in other published research (1) (2).

- We note that this proxy measure is likely to differ in its appropriateness in the US and UK systems, The effect of SES on health care use may be different in the UK compared to the US due to differences in health system structure. In the UK, there is no difference in NHS health services offered on the basis of SES, as all care, both basic and specialised, is free at the point of use (with some small exceptions, e.g. some prescription charges).

- The group of services that we have found is associated with internet use, in the UK setting, are part private and part public, in the sense that some have co-payments (dentists, opticians) or are private in parallel with public (chiropodists, counsellors and physiotherapists), and some are wholly private (chiropractors) or wholly public (emergency services and smoking cessation services). Use of the internet could be implicated in a person’s ability to find any of these community-based services, except perhaps the emergency services.

- We have added in further clarification and discussion of this point in the Methods section (see page 6-7, Covariates paragraph) and the Discussion section (see page 12, first paragraph of Discussion section).

For any relationship study, the selection of covariates was determined by the study purpose, and each covariate needs clear rationale for being included in the model. Most covariates were included without any explanation, for example, loneliness scale, season at start. Further, why SF24 was used and why it was broken down to two scores?

- We thank the reviewer for highlighting this. We have added justification for inclusion of each covariate, and we have explained more clearly that the SF-12 is commonly assessed and reported as its two component scores – mental and physical (see Methods, top of page 7).

One of the outcome variables was “wash/meals assistance”, an important indicator of disability. SF-12 physical score is a good measure of physical health or disability, so no surprise that it’s a strong predictor of using wash/meals assistance. Therefore to examine the relationship between frequent use and “wash/meals assistance” is less meaningful without explaining the relationship between disability and wash/meals assistance.

- In the UK context, ‘wash/meal assistance’ is important as an indicator that the person has a disability serious enough to need help with personal care. For many older people their health (physical or mental) may limit the activities listed within the SF-12 (e.g. vacuuming, playing golf, climbing several flights of stairs) but a much smaller group needs personal care related to keeping clean and assistance with food. This variable therefore identifies a smaller more impaired sub-group. We feel it is valuable therefore to explore the relationship between internet use and this type of care.

- We have added a sentence to the discussion to clarify this point – we would expect this group to have some degree of disability and therefore also potentially less internet use, but we did not demonstrate this in our study (see page 12, second paragraph).

Other comments:

The Introduction should go straight to the purpose of the study and build up the significance and rationale of the study. The Introduction has limited literature on digital divide and health disparities but includes lots of information about MRA-O and WISH, it appears the study was driven by the data from WISH not the other way around.

- We thank the reviewer for this observation. We have amended the Introduction section and included more detail on the digital divide (see page 4, third and fourth paragraphs).

The Methods should follow the format of published articles and detail data sources, sampling and collection methods. If scales were used, Cronbach alpha should be reported.

- We thank the reviewer for this observation. We have added some extra elements to the Methods section to reflect these helpful suggestions (see page 5).

The Data Analysis section should describe whether hierarchical modelling was used given that data were collected from multi-stage sampling strategy and longitudinal design. Why GP surgery contribution was used as random effect needs explanation too. How missing data were handled needs to be included in Methods. If only complete cases were included in data analysis, the actual sample size should be reported.

- We thank the reviewer for these suggestions; we have incorporated them into the relevant sections and given explanations where required (see page 7, Analysis and Missing data sections, and Table 2).

Results: the tables should be formatted following examples of published articles; usually % is sufficient when describing sample characteristics. When reporting relationship (either binary or multivariate) it's a common practice to indicate p-value.

- We thank the reviewer for this suggestion, and we have changed the formatting of the tables to reflect these points. Where the denominator is not the same throughout a column/row, we have indicated how many missing values there are.

Discussion: it's important to discuss the study within the existing literature, especially related studies on older adult's digital divide and health disparities.

- We agree that this would strengthen the paper and have added further discussion to take these points into account (see page 12, last two paragraphs; and page 13, first two paragraphs).

References

1. Solé-Auró A, Alcañiz M. Educational attainment, gender and health inequalities among older adults in Catalonia (Spain). *International Journal for Equity in Health*. 2016; 15: p. 126.
2. Jankovic S, Stojisavljevic D, Jankovic J, Eric M, Marinkovic J. Association of socioeconomic status measured by education, and cardiovascular health: a population-based cross-sectional study. *BMJ Open*. 2014; 4: p. e005222.

VERSION 2 – REVIEW

REVIEWER	Dr Pippa Burns University of Wollongong, Australia
REVIEW RETURNED	28-Apr-2017

GENERAL COMMENTS	The authors should be congratulated on adequately addressing the concerns raised in the first round of review.
--

REVIEWER	Y. Alicia Hong
-----------------	----------------

	Texas A&M University, USA
REVIEW RETURNED	16-Apr-2017

GENERAL COMMENTS	The revised manuscript has addressed most of the comments from the reviewers. It's in a better shape. I have the following comments to further improve the manuscript.  1. Sampling. The author stated the study was based on a "random sample". How were the older adults from 5 GP in 2 regions of southern England were selected? If it's stratified representative sampling, multi-level analysis controlling for weight should be applied in data analysis. If it's simply random sampling, how was it accomplished? 2. Representation of sample. Only 526 of the 1,550 older adults responded the letter, therefore, the response rate was only 30%, and 454 participants returned M-RAO, even less than 30%. There was obvious volunteer bias and self-selection bias in the participants. This major weakness of no representativeness should be noted in the manuscript. 3. One of the strengths the authors reported was "lack of bias", as "the question on internet use was one of many, so participants will not have attached much weight to it and should minimize any reporting bias". Such statement was not true, reporting bias exists in any survey research, regardless how many questions were included in the survey. 4. Introduction should be more focused. For example, the first couple of sentences were too generic and not necessary. The authors should also provide statistics of Internet use in England, and in older adults in England, so readers can have a better understanding of the social context of the current study. 5. The human subjects protection should be reported. How participants gave consent, were they compensated, was the study approved by the institute review board (or other human subjects protection agency) should be reported. 6. The Cronbach alpha of scales used in the manuscript, including loneliness, social isolation, SF-12 should be reported, as indicated in previous review. 7. Table 1 should report significance level on the relationship of Internet use and covariates, use chi-square to analyze the relationship between categorical variables. 8. Table 2 is not necessary, the data missing should be reported in Data Analysis section. The distribution (%) of using healthcare services can be added to Table 1, in which their binary relationship with Internet use can also be demonstrated. 9. Table 3 on binary relationship is not necessary as the information can be presented in Table 1. 10. Discussion should including comparison of findings from the current study and other studies.
--

VERSION 2 – AUTHOR RESPONSE

Reviewer(s)' Comments to Author:

Reviewer: 2

Y. Alicia Hong

Texas A&M University, USA

Please state any competing interests or state 'None declared': None declared.

Please leave your comments for the authors below

The revised manuscript has addressed most of the comments from the reviewers. It's in a better shape. I have the following comments to further improve the manuscript.

1. Sampling. The author stated the study was based on a "random sample". How were the older adults from 5 GP in 2 regions of southern England were selected? If it's stratified representative sampling, multi-level analysis controlling for weight should be applied in data analysis. If it's simply random sampling, how was it accomplished?

- We thank the reviewer for this comment. The sampling strategy used in the study was simple random sampling of all registered eligible patients. It was completed by the participating practices using their electronic records systems, which can generate random sample of their list of patients. We have included comments to this effect in the Participants section (page 5).

2. Representation of sample. Only 526 of the 1,550 older adults responded the letter, therefore, the response rate was only 30%, and 454 participants returned M-RAO, even less than 30%. There was obvious volunteer bias and self-selection bias in the participants. This major weakness of no representativeness should be noted in the manuscript.

- We thank the reviewer for this comment, we agree that this is a limitation and we have included this point in the revised manuscript in the penultimate paragraph of the Discussion (page 14).

3. One of the strengths the authors reported was "lack of bias", as "the question on internet use was one of many, so participants will not have attached much weight to it and should minimize any reporting bias". Such statement was not true, reporting bias exists in any survey research, regardless how many questions were included in the survey.

- We thank the reviewer for this comment. What we meant by this statement was that a survey that focuses on internet use in later life may introduce some selection bias in that those more interested in the internet may be more likely to participate, but this is not the case with our study as a general health and well-being survey. We have changed this Strength to a more general point (page 3).

4. Introduction should be more focused. For example, the first couple of sentences were too generic and not necessary. The authors should also provide statistics of Internet use in England, and in older adults in England, so readers can have a better understanding of the social context of the current study.

- We thank the reviewer for these comments. Regarding the second point above, we have added further discussion of current trends of internet use in this population to the Introduction (page 4).

- Regarding the first two sentences of the Introduction, we agree that they begin broadly, but we consider that introducing the topic is important, as this journal discusses a wide variety of topics. Also, we wish to give an outline of the context in which this work has taken place, and we feel that this is an appropriate place for this work to be mentioned. We also note that a previous reviewer had requested inclusion of additional literature on the digital divide and health disparities, which is another reason why we would like to have this information here.

5. The human subjects protection should be reported. How participants gave consent, were they compensated, was the study approved by the institute review board (or other human subjects protection agency) should be reported.

- We thank the reviewer for this comment, and we have added a comment on informed consent in the Introduction (page 4-5). We give the London-East Research Ethics Committee reference number in the Funding section of this manuscript on page 3. No financial incentive was given to patients.

6. The Cronbach alpha of scales used in the manuscript, including loneliness, social isolation, SF-12 should be reported, as indicated in previous review.

- We thank the reviewer for this comment. No Cronbach's alpha was calculated as part of this study. These scales are well-established commonly used scales. Published Cronbach alphas exist of all these scales (and all report high internal consistency with Cronbach alphas of >0.7), however, we feel that giving enough information within the paper for the reader to understand the Cronbach's alphas from the published literature would detract from the main study.

7. Table 1 should report significance level on the relationship of Internet use and covariates, use chi-square to analyze the relationship between categorical variables.

- We thank the reviewer for this comment. Table 1 is not intended to give results of the regression analyses; it is instead the presentation of baseline characteristics of the patients, and those characteristics are given for the overall group of participants as well as for each internet group (frequently and infrequently). We have edited the legend to Table 1 to clarify this.

- Tables 3 and 4 show the results of the panel regression analyses, using the panel dataset, i.e. not just baseline data. These analyses looked at the resource use at baseline, 3 months and 6 months as a function of baseline internet use, controlling for the other baseline variables listed. The chi-squared test was used for Age bands (the categorical variable) and this is presented in Tables 3 and 4 (also see Methods section on page 7).

- We feel that mixing up Table 1 and Table 3 could give rise to some confusion around the distinction between baseline patient characteristics and the panel nature of the regression analysis so we have not combined the tables. We have however made modifications to the Results section (page 8-9) to show this more clearly. We have also modified the legends of Tables 3 and 4.

8. Table 2 is not necessary, the data missing should be reported in Data Analysis section. The distribution (%) of using healthcare services can be added to Table 1, in which their binary relationship with Internet use can also be demonstrated.

- We thank the reviewer for this comment. Table 2 shows the missing resource use variables at each of the three timepoints. Table 1 only considers fixed patient characteristics, measured at baseline. We consider that the merging of these two tables would be confusing given these differences, so we have opted to keep them as they are. However, we note the reviewer's helpful highlighting of how we could better represent the data in the Tables by making the legends clearer, and we have done this by modifying the legends of Tables 1 and 2, as well as moving and modifying some of the text in the Results section (page 8-9).

9. Table 3 on binary relationship is not necessary as the information can be presented in Table 1.

- We thank the reviewer for this comment. We agree that there are some similarities between the aspect of the information presented in Tables 1 and 3, but again there is the consideration that Table 1 lists baseline proportions and numbers of missing values, and if we were to add that to Table 3 it might suggest that these levels of missingness were the same across all timepoints, which is not the case. For this reason we consider that it is clearer to keep these two tables separate.

10. Discussion should including comparison of findings from the current study and other studies.

- We thank the reviewer for this comment. We have added further discussion comparing the findings from our study with other work in the Discussion section (page 13-14).

Reviewer: 1

Dr Pippa Burns

University of Wollongong, Australia

Please state any competing interests or state 'None declared': None

Please leave your comments for the authors below

The authors should be congratulated on adequately addressing the concerns raised in the first round of review.

- Thanks very much!

VERSION 3 – REVIEW

REVIEWER	Y. Alicia Hong Texas A&M University
REVIEW RETURNED	02-Jun-2017

GENERAL COMMENTS	The authors have addressed all the reviewers' comments and the manuscript is ready for publication.
---